# Predicting Emergent Capabilities by Finetuning

**Charlie Snell**      **Eric Wallace**      **Dan Klein**      **Sergey Levine**
University of California, Berkeley

## Abstract

A fundamental open challenge in modern LLM scaling is the lack of understanding around emergent capabilities. In particular, language model pretraining loss is known to be highly predictable as a function of compute. However, downstream capabilities are far less predictable—sometimes even exhibiting emergent jumps—which makes it challenging to anticipate the capabilities of future models. In this work, we first pose the task of emergence prediction: given access to current LLMs that have random few-shot accuracy on a task, can we predict whether future models (GPT-N+1) will have non-trivial accuracy on that task? We then discover a simple insight for this problem: *directly finetuning LLMs on a given task can shift the point in scaling at which emergence occurs towards less capable models*. To operationalize this insight, we can finetune LLMs with varying amounts of data and fit a parametric function that predicts when emergence will occur (i.e., "emergence laws"). To validate this approach, we use four standard NLP benchmarks where large-scale open-source LLMs already demonstrate emergence (MMLU, GSM8K, CommonsenseQA, and CoLA). Using only small-scale LLMs, we find that, in some cases, we are able to accurately predict whether models trained with up to 4x more compute have emerged.

## 1 Introduction

The pretraining loss for language models has been shown to follow a simple predictable power law as a function of compute, model parameters, and data (Kaplan et al., 2020; Hoffmann et al., 2022). This finding has enabled a precise empirical science to develop around the scaling behavior of LMs (Muennighoff et al., 2023; Aghajanyan et al., 2023; Krajewski et al., 2024), which has in turn led to much of the rapid improvement of language model capabilities in recent years (OpenAI et al., 2024; Team et al., 2023). On the other hand, the specific downstream capabilities corresponding to a given pretraining loss are generally much less predictable, posing significant challenges for 1) model developers who may want to make specific model or architectural decisions on the basis of future LM capabilities; 2) policymakers who will need time to assess, plan for, and prepare for potentially dangerous future LM capabilities like deception, bio-risk, or malicious software agents (Shevlane et al., 2023; Bengio et al., 2023); and 3) the ability for stakeholders to make reliable business, financial, and investment decisions on the basis of future LLM capabilities.

Of particular concern and interest is the phenomenon of emergent capabilities in large language models Wei et al. (2022a). On certain downstream tasks, models may exhibit "emergence" wherein only beyond a certain, seemingly arbitrary, threshold in LM scaling (e.g. the point of emergence) do models spontaneously improve beyond random-chance. In cases where existing models have already crossed the point of emergence on a given task, demonstrating smooth performance improvements (the post-emergence regime), it is possible to make highly accurate predictions about the performance of future models (Caballero et al., 2023; Gadre et al., 2024; OpenAI et al., 2024; Owen, 2024). However, on tasks in which all existing models demonstrate random-chance performance (the pre-emergence regime), making any kind of prediction at all about future model capabilities remains an important unsolved challenge in LM scaling. In this case, there is no known method for predicting

---

Corresponding author(s): csnell22@berkeley.edu

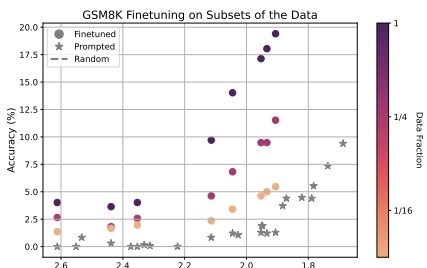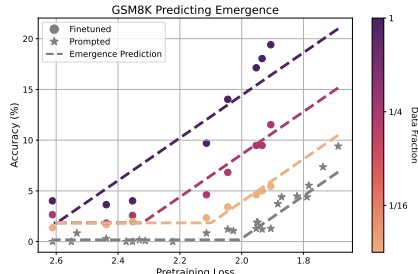

Figure 1: On the left, we compare LLMs that are few-shot prompted or finetuned on GSM8K. Our key finding is that *finetuning effectively shifts the point of emergence from stronger to weaker models*. Moreover, by varying the amount of finetuning data, the emergence point is shifted accordingly. We can leverage this to predict when few-shot emergence will happen by fitting parametric functions on the emergence points from finetuning and taking the limit (right figure).

at what point, if any, models will demonstrate emergence, let alone how performance will scale thereafter. To this end, we pose the problem of emergence prediction. Concretely, can we accurately predict the point in scaling at which emergence will occur on a given task, while only having access to pre-emergence model checkpoints?

We demonstrate an extremely simple yet highly effective solution to this problem; namely that it is possible to predict fewshot emergent capabilities in future LMs (e.g. GPT-N+1) by finetuning today's weaker LMs (e.g. GPT-N). For example, in Figure 1 left, we see that finetuning models for GSM8K, rather than prompting them, systematically shifts the point of emergence on this task from stronger to weaker LMs. Moreover, by varying the amount of finetuning data, the emergence point is shifted accordingly. Motivated by this finding, we develop an emergence law; a parametric function that models how the point of emergence shifts as a function of the size of the finetuning dataset. Using this emergence law, we can then extrapolate a prediction of the point in scaling at which emergence will occur in the few-shot setting (Figure 1 right).

To validate our approach, we use four standard NLP benchmarks—MMLU, GSM8K, CommonsenseQA, and CoLA—where large-scale open-source LMs have already demonstrated emergence. By fitting an emergence law using only small-scale, pre-emergence LLMs, we find that we are not only able to accurately predict the point in scaling at which emergence will occur with more capable LLMs, but also in some cases, we can do so using models trained with only 1/4th the FLOPS needed to achieve emergence.

## 2 Background

**Emergence in Large Language Models.** Emergence Wei et al. (2022a) refers to the phenomenon in which past a certain point in LM scaling models suddenly improve beyond random-chance performance on a given task. As a result, despite the fact that pretraining loss can be reliably predicted as a function of model scale, the downstream capabilities corresponding to a particular loss cannot. In our work, we define the x-axis for emergence ("model scale") to be the pretraining loss, rather than the number of model parameters or the FLOPS used in pretraining Wei et al. (2022a). In line with recent works (Du et al., 2024; Gadre et al., 2024; Huang et al., 2024; Xia et al., 2023), we find that the pretraining loss is highly predictive of downstream capabilities, and thus makes for a more natural and precise independent variable when studying emergence.

**Emergence is a Mirage?** A recent work by Schaeffer et al. (2023) claimed that the phenomenon of emergence observed by Wei et al. (2022a) is not due to sudden fundamental changes in the model, but rather due to the researcher's choice of metric. Namely, they argue that emergence is typically observed when using discontinuous metrics of accuracy, such as

exact-match, and that using a continuous metric, such as the model's correct answer probability, will instead yield smooth and predictable improvements in performance. However, in many practical policy-relevant and realistic settings, the metric of primary interest may be fundamentally discontinuous. Additionally, in some cases finding metrics that yield smooth scaling behavior may be challenging (Barak, 2023). In particular, in Figure 10 we show that when using a continuous probability-based evaluation metric on two canonical language model benchmarks, namely MMLU (Hendrycks et al., 2021) and CommonsenseQA (Talmor et al., 2019), we still observe emergence. Therefore, additional tools are needed in order to resolve the practical challenges induced by the phenomenon of emergence.

**Practical challenges with emergence.** The phenomenon of emergent capabilities introduces significant challenges for safety preparedness, frontier model development, and business decision making with regards to language models. In particular, it is possible that future language models may demonstrate dangerous emergent capabilities, such as planning, deception, bio-risk, or the ability to generate malicious software (Anwar et al., 2024; Hendrycks et al., 2023). Without any ability to predict when and if these capabilities will emerge, we are only left to speculate, which is highly sub-optimal given the potentially high stakes. Furthermore, model developers may need to make architecture and dataset decisions on the basis of downstream capabilities which may only emerge with scale, making it challenging to do so. As a result, the inability to reason about and predict emergent capabilities in advance presents a significant unsolved challenge in language model scaling.

## 3 Scaling Laws for Emergence Prediction

We now introduce the problem of emergence prediction, and then we describe our specific approach for predicting emergence.

### 3.1 Emergence Prediction

We define emergence prediction as the problem of identifying the point in scaling at which emergence will occur using only checkpoints from models that are pre-emergence (i.e., have near-random or trivial performance on a task of interest). In our case, we define "model scale" to be the pretraining loss of a given model. We therefore aim to predict the point in the pretraining loss at which emergence will occur.

Making an exact pointwise prediction may not always be reasonable, and indeed due to noise in both the model checkpoints and the prediction method, we may have considerable uncertainty about the exact point of emergence. To this end, we not only aim to predict a single maximum likelihood point-wise estimate for when emergence will occur, but also a calibrated probability distribution over all possible points at which it may occur.

### 3.2 Finetuning shifts the point of emergence

Our specific approach to emergence prediction builds off of the observation that on tasks which exhibit emergence, finetuning models for a specific task can effectively shift the point of emergence from strong to weak models. In particular, we see that in Figure 1 and Figure 2 that finetuning models on MMLU, GSM8K, and CommonsenseQA significantly shifts the point at which emergence would typically occur in the few-shot setting towards less capable models. Furthermore, by varying the size of the finetuning dataset, the magnitude of this shift is adjusted accordingly.

To determine the extent to which a similar shift in emergence can be achieved by merely tuning the few-shot prompt, in Appendix A.1, we conduct experiments with continuous prefix tuning (Li & Liang, 2021) and with varying the number of shots in the prompt. We find that neither of these strategies are able to produce a meaningful shift in the point of emergence. On the other hand, in Appendix A.1, we find that low-rank finetuning (e.g. LoRA) (Hu et al., 2021) shifts the point of emergence to nearly the same degree as that of full parameter finetuning, suggesting that modifying model parameters may be critical for

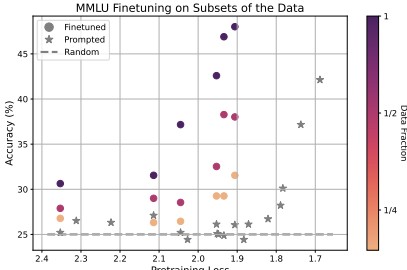 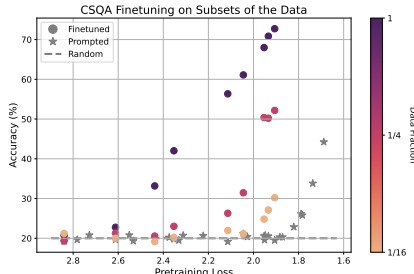

Figure 2: On both MMLU and CommonsenseQA, we observe a clear trend in which finetuning effectively shifts the point of emergence from stronger to weaker models. Moreover, the size of the finetuning dataset controls the magnitude of this shift.

inducing an emergence shift. However, we leave further exploration of this phenomenon to future work.

Taken together, these results indicate that finetuning language models for specific tasks can unlock capabilities that are not inherently present in the base model itself, and thus, in a sense, weaker models can be made to simulate the capabilities of stronger models. We can then use this insight to precisely predict the exact point in scaling at which these more capable models will demonstrate emergence, by extrapolating how the emergence point shifts as we vary the amount of finetuning data.

## 3.3   Fitting Emergence Laws

To predict when emergence will occur, we can model how the point of emergence shifts as a function of the amount of finetuning data and then take the limit as the amount of finetuning data approaches the number of examples used in the few-shot setting. In particular, we develop an *emergence law*: a simple scaling-law inspired parametric function from which we can precisely extrapolate the point in scaling at which emergence will occur in the few-shot setting. In this section, we will first describe our procedure for collecting the data to fit our emergence law, and then we will describe the specific functional form that we use.

**Collecting datapoints for emergence modeling.**   Our emergence law takes as input two variables: the pretraining loss corresponding to a particular model and the amount of data used for finetuning it. It then outputs a prediction of the performance of the finetuned model on our task of interest. The parameters of our emergence law are fit to a number of empirical examples of this form obtained by finetuning different models on varying amounts of data. In order to accurately model the emergence, we will therefore need 1) a sufficiently granular set of models of varying capability levels that were pretrained on the same data, and 2) a wide range of finetuning data amounts.

To satisfy 1), we can avoid the need to fully pretrain multiple models from scratch by instead utilizing a number of evenly spaced intermediate checkpoints from a single pretraining run. Similar to recent work (Du et al., 2024; Gadre et al., 2024), we find that downstream performance is highly correlated with pretraining loss, and thus using multiple intermediate checkpoints from the same training run is just as effective as using separate pretraining runs of varying scale.

To obtain a wide range of finetuning data amounts, we take random subsets of the full finetuning dataset for a given task. Since we ultimately want to predict emergence by extrapolating our model into the low-data limit, it is important that we focus our experiments on smaller data amounts for which we can still observe emergence with weaker models. In our case this is often on the order of a few hundred finetuning examples.

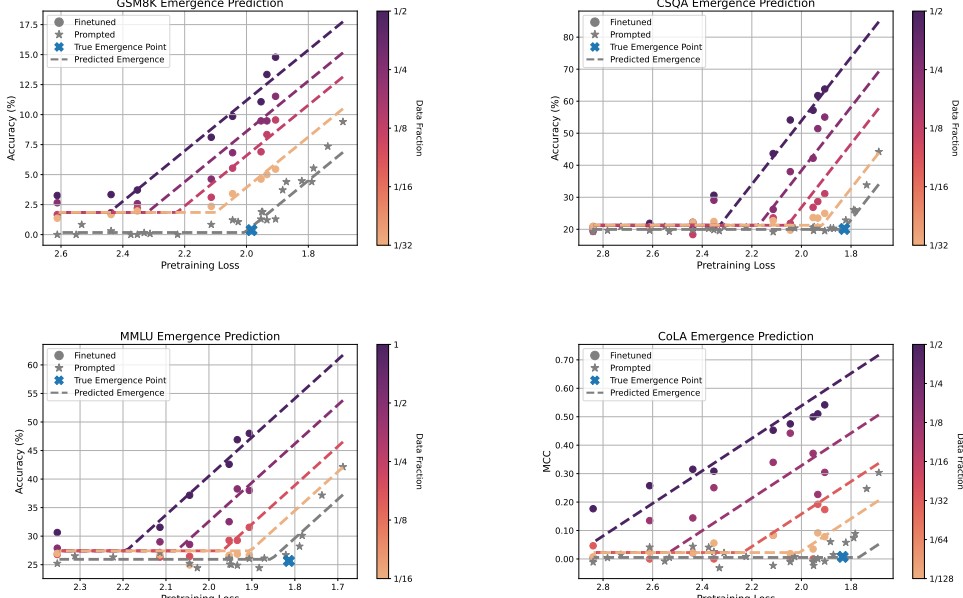

Figure 3: We plot the maximum likelihood predictions from our emergence law on each task. On each plot we include a subset of the finetuning datapoints used for fitting the emergence law (see Appendix A.5 for all the datapoints). The grey line represents our extrapolated prediction and the multi-color lines correspond to the fit produced by the emergence law for the various data levels. We see that across all tasks we are able to successfully predict the point of emergence within 0.1 nats and in many cases much less than that.

**Modeling the point of emergence.** After finetuning a number of different language models on varying data amounts, we can then fit an emergence law to the results. The emergence law should effectively model how the point of emergence shifts as we vary the amount of finetuning data. We find that this shift is well modeled by a power-law in the logarithm of the amount of finetuning data:

$$E(D) = k * \log(D)^\alpha + C$$

Where $D$ is the number of finetuning examples. We also experimented with using a power-law in data amount (e.g. without the logarithm) but found this to generally produce worse extrapolations (see Appendix A.4). To then model the actual emergence in the downstream performance, we use a ReLU, where the elbow of the ReLU is parameterized by $E(D)$:

$$\text{Perf}(L, D) = A * \max(E(D) - L, 0) + B$$

Here L is pretraining loss (in nats). While prior work Gadre et al. (2024) has found downstream performance to be well modeled by an exponential in pretraining loss, we use a ReLU in this work because our primary focus is on modeling the point at which emergence occurs, rather than accurately extrapolating performance after emergence. The ReLU elbow, in this case, provides a clear denotation of the precise point at which emergence begins.

In addition to the results from our finetuning runs, we may also optionally want to include the few-shot results from our pre-emergence models when fitting the emergence law. While these few-shot results should all have near-random performance, they will still be somewhat informative for telling the model that emergence hasn't happened yet.

To effectively enable this, we add an optional additional parameter $\Delta$ which effectively models the upwards shift in the base of the ReLU that we see when finetuning in Figure 2

and Figure 1. We believe that this shift is likely due to the model learning trivial features from the fine-tuning data, such as the base-rate of the correct answer [1]. Our final model is:

$$\text{Perf}(L, D, \mathbb{1}_{\text{is finetuned}}) = A * \max(E(D) - L, 0) + B + \Delta * \mathbb{1}_{\text{is finetuned}}$$

In total our emergence law has 5 parameters – $A$, $B$, $k$, $\alpha$, $C$ – and an optional 6th parameter $\Delta$. To obtain an maximum likelihood estimate for the point of emergence, we fit these parameters to our data, and then we take the limit under $E(D)$ as $D$ approaches the number of examples in our fewshot prompt. We can also obtain a probability distribution for the point of emergence by running MCMC on the posterior of the emergence law parameters.

To learn the parameters, we use mean squared error and weight the loss for each datapoint in an inverse proportion to the size of the finetuning dataset. Intuitively this ensures that the optimizer focuses on fitting the emergence more accurately in the low data limit, since this is where we ultimately want to extrapolate.

## 4 Predicting Emergence

We now empirically validate the efficacy of our emergence law, on standard NLP benchmarks where large-scale open-source LLMs already demonstrate emergence. In this setting, we fit an emergence law using smaller-scale LLMs, which have random chance performance on the task, and then check the accuracy of our predictions against the true point of emergence observed with larger models. We are particularly interested in understanding exactly how far in advance, in terms of pretraining FLOPS, our emergence law can successfully predict the point of emergence.

**Models and tasks.** Since fitting our emergence law requires access to a number of intermediate model checkpoints, we use the OpenLLaMA V1 models Geng & Liu (2023), for which we have checkpoints for models of three different sizes: 3B, 7B, and 13B. We also include an additional experiment with Open-LLaMA V2 in Appendix A.2. We use 4 standard NLP benchmarks—MMLU (Hendrycks et al., 2021), GSM8K (Cobbe et al., 2021), CommonsenseQA (Talmor et al., 2019) (CSQA), and CoLA (Wang et al., 2018)—for which we observe emergence in the few-shot setting with our most capable model checkpoints. On each of these tasks, we find that all 3B model checkpoints have yet to emerge, whereas the 7B and 13B models demonstrate emergence by the end of training. We therefore only fine-tune the 3B model checkpoints for fitting our emergence law, and then make use of the 7B and 13B checkpoints for validating the accuracy of our predictions. We include additional details in Appendix A.6.

**Fitting the emergence law.** We fit the emergence law parameters with the L-BFGS optimizer, following the procedure in Hoffmann et al. (2022). We obtain a distribution for the point of emergence by taking 100k posterior samples using the No-U-Turn Sampler (Hoffman et al., 2014). The ground truth point of emergence is determined by fitting a ReLU to the full set of few-shot results of all model checkpoints. We consider a prediction successful if the predicted emergence point of our maximum likelihood estimate falls within 0.1 nats of the true emergence point. Additional details are in Appendix A.7.

**Can we successfully predict emergence?** In Figure 3, we plot the maximum likelihood fit for our emergence law on all four tasks. We see that on all tasks our emergence law our predictions of when emergence will happen are overall very accurate and fall well within 0.1 nats of the true emergence point[2]. In Figure 4, we additionally plot the cumulative distribution function (CDF) of our MCMC posterior estimate for the emergence point. We

---

[1]For example, on GSM8K simply always guessing the most common answer in the training set achieves 2.6% accuracy on the test-set.

[2]We note that on GSM8K three of the 3B model checkpoints that we used for fitting the emergence law are technically considered post-emergence by the ground truth. However, these points have trivial fewshot performance on the task $< 3\%$, and therefore without access to the more capable models they

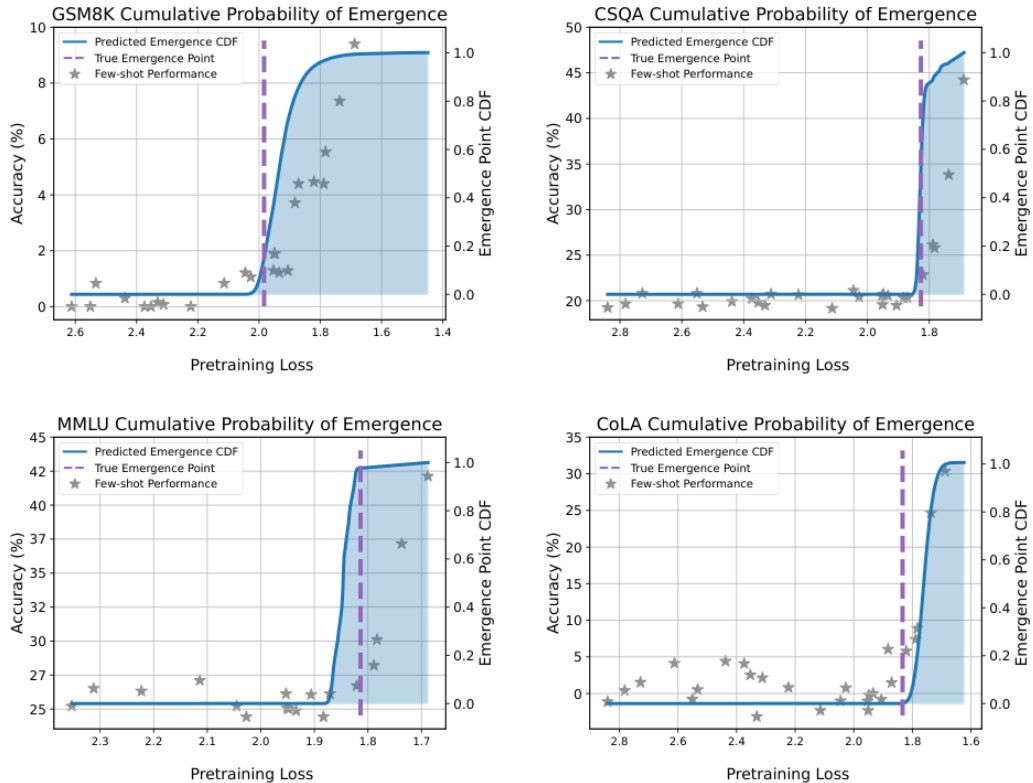

Figure 4: We plot the cumulative distribution function of our estimated posterior distribution over the point of emergence on each task. The stars correspond to few-shot performance on the task and represent the true emergence curve. The point at which the slope of the CDF peaks represents the mode of the distribution. We see that across all tasks that the distribution spikes near the true point of emergence and is followed by a long tail.

see a general trend in which the distribution spikes near the true emergence point, followed by a somewhat long tail: our emergence law is generally confident about when emergence will occur but has some uncertainty about the possibility of emergence occurring much later.

**How far in advance can we predict emergence?** Now that we've demonstrated that we can reliably predict when emergence will occur using our pre-emergence 3B model checkpoints, we would like to better understand exactly how far in advance, in terms of pretraining FLOPS, we can successfully make such predictions. To do this we can hold out additional 3B checkpoints when fitting our emergence law. In Figure 5, we plot our emergence predictions against the number of pretraining FLOPS required for training the most capable model used for emergence prediction. We find that the degree to which we can predict emergence in advance is somewhat task dependent. In particular, on GSM8K and MMLU we are able to reliably make predictions well in advance of the emergence point. On CommonsenseQA and CoLA, on the other hand, we find that our ability to make predictions in advance is much more limited. In particular, if we compare the minimum FLOPS from which we can make a successful emergence prediction against the FLOPS required for training the first post-emergence checkpoint, we can obtain an estimate for how many FLOPS in advance our emergence law can make predictions. We find that on MMLU and GSM8K we can successfully predict emergence up to 4.3x and 3.9x FLOPS in advance respectively. However, on CommonsenseQA and CoLA we are only able predict by a factor of 1.9x and 2.3x in advance respectively.

would generally be considered pre-emergence. Additionally, in Figure 5, we find that even in the absence of these three checkpoints, we are still able to make successful predictions.

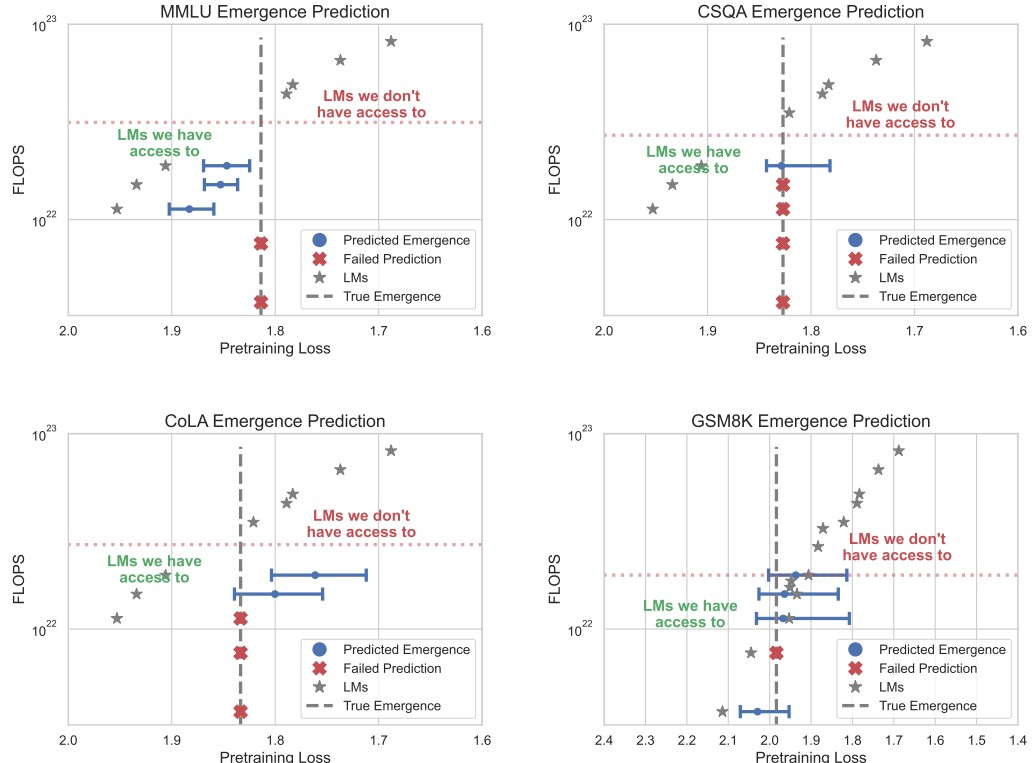

Figure 5: On each task we hold out model checkpoints to understand how far in advance, in terms of pretraining FLOPS, we can successfully predict emergence[5]. The y-axis of each blue error-bar represents the maximum number of pretraining FLOPS used for fitting the emergence law. Each blue circle represents the median of the emergence prediction posterior, and the error bar represents the range of the 5th to 95th percentile predictions. If the maximum likelihood estimate for an emergence is further than 0.1 nats from the true emergence point, we consider the prediction unsuccessful and denote it by a red-cross on the plot[6]. "LMs we have access to" refers to the set of pre-emergence model checkpoints which we use for fitting our emergence law. "LMs we don't have access to" refers to our held-out 7B and 13B OpenLLaMA V1 checkpoints. We see that on MMLU we are able to make reasonable predictions about the point of emergence using models trained with $\sim 10^{22}$ FLOPS, but not with fewer than $10^{22}$ FLOPS. The first post emergence model on MMLU was trained with $\sim 5 * 10^{22}$ FLOPS, hence we are able to predict 4-5x the FLOPS in advance on this task. For GSM8K, we can do the same calculation, and we see that in this case we are also able to predict emergence with up to 4x the FLOPS in advance. However, on CoLA and CommonsenseQA (CSQA) we are only able to predict emergence 2x the FLOPS in advance.

## 5 Limitations and Future Directions

We found that our specific approach to emergence prediction (e.g. emergence law) is able to accurately predict the point of emergence with up to 4x the FLOPS in advance, which represents meaningful progress on the challenging unsolved problem of emergence prediction. However, our prediction ability falls far short of the 1000x demonstrated in OpenAI et al. (2024) for predicting post-emergence downstream capabilities. Such advance predictions may be necessary in some high-stakes settings; therefore further progress on the problem of emergence prediction is needed. Below we outline the limitations of our approach and potential future directions for improving emergence prediction.

**Better data subset selection may improve predictions.** In this work, we did not specifically optimize our experimental setup for maximizing the degree to which we can make predictions in advance. In particular, in Figure 5 we use the same data subsets for making predictions at all FLOPS levels. By training on more data subsets specifically chosen to be closer to the limit at which emergence with finetuning is observable, we may have been able to further improve our predictions at earlier FLOPS levels. Additionally, all of the finetuning datasets we used have no more than 10K examples in total, which we found to generally be sufficient for our experiments. However, it is possible that using larger finetuning datasets (e.g. on the order of 100K or more examples) may enable even larger shifts in the emergence point, allowing for even further advance capability prediction. We believe that these limitations in our data subset selection procedure may be the reason why on some tasks we were able to predict 4x the FLOPs in advance and on others only 2x. Though we leave further exploration of this to future work.

**What methods can shift the point of emergence?** In Section 3.2 we showed that targeted finetuning can systematically shift the point of emergence towards less capable models. We also experimented with alternative methods of shifting the point of emergence, finding that modifying model parameters seems to be critical for effectively shifting the point of emergence. However, our understanding of this phenomenon is limited, and future work should aim to better understand the limitations of and mechanisms by which the point of emergence can be shifted.

**Predictions may not transfer in some settings.** All of our experiments are conducted using a series of transformer model checkpoints that were pretrained on the same corpus and that only meaningfully differ in the amount of data used for pretraining and their parameter counts. LMs with sufficiently different architectural choices (e.g. state-space models) (Tay et al., 2022; Arora et al., 2023; Gu & Dao, 2023) or trained on a different pretraining corpa may exhibit different downstream capabilities even at the same level of pretraining loss. Our emergence laws may therefore not successfully transfer to extrapolating predictions in these settings.

**Task-specific finetuning is limited.** Prior work Gudibande et al. (2023) has shown that finetuning models broadly on many tasks at once, often fails to significantly improve the model's capabilities. In our work, we instead focus on finetuning models for specific tasks. However, it is possible that for certain more complex capabilities of interest, models may be required to compose many different skills together. In these settings, finetuning may be a less effective tool for enabling emergence prediction.

## 6 Related Work

**Emergence in LLMs.** A number of works Wei et al. (2022a); Suzgun et al. (2022); Ganguli et al. (2022) have observed the phenomenon of emergence in language models, wherein certain downstream capabilities suddenly appear beyond a certain model scale. However, more recent works Schaeffer et al. (2023); Xia et al. (2023); Hu et al. (2024) have suggested that the phenomenon of emergence may vanish when using continuous metrics, in place of discontinuous ones. The generality of this claim has since been challenged by Du et al. (2024), which shows that in some cases emergence can even occur in the presence of continuous metrics. Several works have aimed to provide explanations for why emergence occurs, attributing the phenomenon to in-context learning (Lu et al., 2024), emergent multiple-choice circuits (Lieberum et al., 2023), or the composition of multiple skills (Arora & Goyal, 2023; Hu et al., 2024). Our work is most closely related to concurrent work from Blakeney et al. (2024), which demonstrates that by up-sampling domain-specific datasets at the end of

---

[5]Our emergence prediction is in terms of loss but the quantity of interest is FLOPS. Therefore, we plot pretraining loss on the x-axis and pretraining FLOPS on the y-axis, since there is not a standard 1-to-1 mapping between loss and FLOPS.

[6]We do this because in some cases the failed predictions would be well off the plot and we would like to the keep the axis bounds constrained for presentation clarity.

pretraining, downstream LM performance can be made much more predictable at low FLOP budgets, even with emergent downstream tasks. In contrast to Blakeney et al. (2024), our work instead focuses on predicting the precise point in scaling at which emergence will occur, rather than the degree of scaling thereafter.

**Neural scaling laws.**    A number of works have studied the scaling behavior of language model pretraining loss Kaplan et al. (2020); Hoffmann et al. (2022); Aghajanyan et al. (2023); Muennighoff et al. (2024); Henighan et al. (2020); Krajewski et al. (2024). Most related to our work are Hernandez et al. (2021) and Isik et al. (2024), which study how language model finetuning scales as a function of model size and finetuning data amount. However, these works critically differ from ours, in that they study performance on settings which are already smoothly improving with scale.

**Scaling laws for downstream metrics.**    In addition to understanding the scaling behavior of upstream metrics, a number of works have proposed methods for modeling the scaling of downstream metrics (Caballero et al., 2023; Gadre et al., 2024; Isik et al., 2024; Owen, 2024; Ruan et al., 2024; Hu et al., 2024). These works assume that the downstream metric of interest is already showing some signs of smooth improvement as a function of model scale (i.e. post-emergence). Our work instead focuses on predicting the point in scaling at which capabilities will begin to emerge, given access to only pre-emergence checkpoints.

## 7    Conclusion

In this work we introduced the problem of emergence prediction; namely, identifying the point in pretraining at which performance on a given task demonstrates emergence using only pre-emergence model checkpoints. We then proposed one such approach to making emergence predictions: we fit a scaling-law inspired parametric function (e.g. an emergence-law) from which we extrapolate an emergence prediction. We finally demonstrated the predictive power of our emergence law using a set of four standard LM benchmarks: MMLU, GSM8K, CommonsenseQA, and CoLA. Using only pre-emergence checkpoints we are able to effectively predict the point of emergence using only 1/4th the FLOPS needed to reach the point of emergence on two of our four tasks. We believe that our findings represent meaningful progress towards better understanding and reasoning about emergent capabilities in large language models.

## Acknowledgements

We thank Jason Wei, Kevin Liu, Jonathan Uesato, and Young Geng for discussion and feedback on earlier drafts of our paper. Charlie Snell is supported by the OpenAI Superalignment Fellowship. This research was supported with Cloud TPUs from Google's TPU Research Cloud (TRC).

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

# A Appendix

## A.1 Alternative Methods for Shifting the Point of Emergence

We are interested in understanding other methods that can shift the point of emergence. We therefore conduct additional experiments on MMLU with 1) varying the number of shots in the prompt, 2) performing continuous prefix tuning Li & Liang (2021), and 3) using low-rank finetuning with LoRA. We see in Figure 6 that both prefix tuning and using few-shots in the few-shot prompt have little effect on shifting the point of emergence. On the other hand, in Figure 7 we see that low-rank finetuning shifts the point of emergence to a comparable degree to that of full finetuning, even in the rank-1 setting.

Together these results suggest that updating the model's parameters may be necessary for shifting the point of emergence. Though further exploration of this phenomenon is needed. In particular, it is possible that using hundreds or thousands of shots in the prompt may enable more of a shift than what we observed (Agarwal et al., 2024).

For the prefix tuning baseline we use a prefix length of 8, parameterized by a 2-layer MLP with input and hidden-dim 512. We train with a learning rate of 3e-4 and keep all other hyperparameters the same as our full finetuning experiments. We found attempts to increase the capacity of the prefix tuning (e.g. using a longer prefix or dropping the MLP and directly tuning the embeddings) to make training more unstable and generally yield worse performance.

For LoRA we use the same finetuning hyper-parameters as full fine-tuning, including the learning rate. To ensure that the LoRA updates are of similar magnitude to full finetuning with the same learning rate, we set the LoRA $\alpha$ hyper-parameter to be equal to the model's hidden dimension. We found this setup to yield the best results with minimal changes between our full finetuning and LoRA setup.

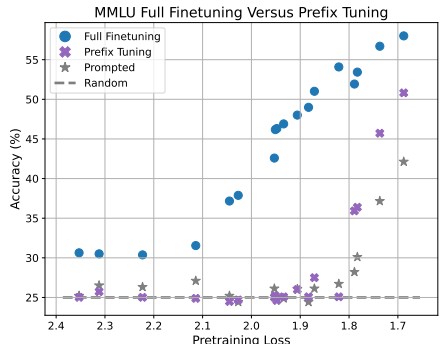 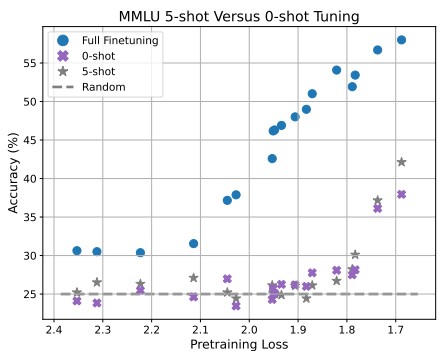

Figure 6: One the left we compare full fine-tuning against continuous prefix tuning on MMLU. We find that prefix tuning provides effectively no shift to the point of emergence, despite improving the performance of post-emergence models. On the right we compare 0-shot verses 5-shot prompting on MMLU. We see that using fewer shots has no meaningful effect on the point of emergence. Together these results suggest that the ability for prompt tuning to shift the point of emergence is very limited.

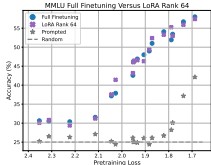 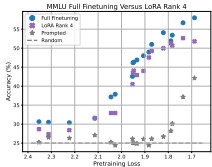 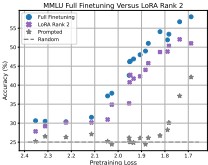 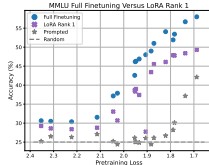

Figure 7: Comparing LoRA finetuning, with rank 1, 2, 4, and 64 against full finetuning on MMLU. We see that LoRA finetuning even with rank 1 shifts the point of emergence to a comparable degree to that of full finetuning.

## A.2    Using Emergence Prediction to Cheaply Make Modeling Decisions

One use case for emergence prediction is to enable model developers to cheaply make modeling decisions on the basis of a model's downstream capabilities. Without emergence prediction, making such decisions may be extremely costly, requiring model developers to train large models before seeing any signal. As we found in Section 4, our emergence law can enable us to use up to 4x less compute when running such experiments.

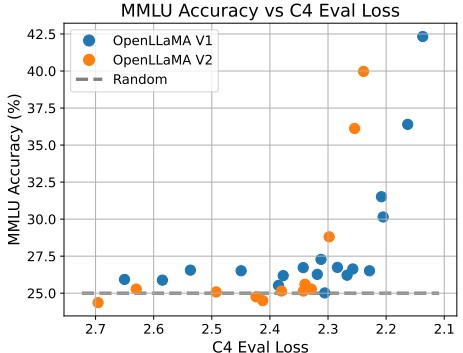 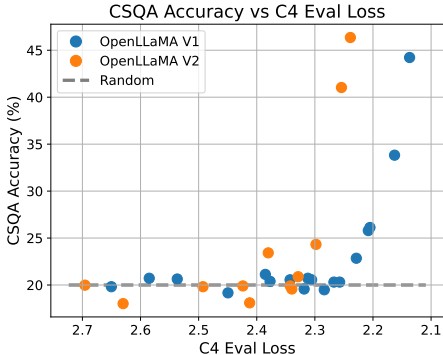

Figure 8: Comparing the point of emergence for few-shot prompted OpenLLaMA V1 and V2 models as a function of C4 validation loss. We see that on both MMLU and CommonsenseQA, the point of emergence differs between the two model series, with V2 emerging earlier than V1 in both cases, suggesting that the pretraining copus used for training the V2 models is likely of higher quality than that used for the V1 models.

**Determining data quality with emergence prediction.**  As a proof of concept of how emergence prediction can be to make modeling decisions, we use emergence prediction to help us determine which of two different pretraining corpa is of higher quality for downstream tasks, using only pre-emergence models. To do this, we extend our experiments with the OpeLLaMA V1 models in Section 4, by also conducting emergence prediction on the OpenLLaMA V2 models (Geng & Liu, 2023) on MMLU, which were pretrained on a different corpus from the V1 models. Since, the two model series (V1 and V2) were trained on different corpa, we should expect them to have different emergence points as a function of loss. We might therefore expect the corpus which shows emergence earlier in pre-training to be the superior dataset for the task at hand. In this way, we can use emergence prediction to help us more cheaply make decisions about pretraining data.

**Comparing V1 and V2 corpa.**  Specifically, the V1 models were trained on the RedPajama (Computer, 2023) dataset, whereas the V2 models were trained on a custom mixture of the Falcon Refined-web (Penedo et al., 2023) dataset, the StarCoder (Li et al., 2023) dataset, and the RedPajama (Computer, 2023) dataset. The V2 models also used a different tokenizer, which was trained on its corresponding pretraining dataset. Most other hyperparameters between the two series of models remain the same.

Since the two model series were pre-trained on different corpa, their respective pretraining losses are not strictly comparable, making the point of emergence in terms of these different x-axes largely uninformative for deciding which dataset is better. In order to enable an informative comparison between the point of emergence for models trained on different corpa, we use loss on a standardized held out corpus as the x-axis. Specifically, we use loss on the C4 validation set. In Figure 8, we plot fewshot performance as a function of C4 validation loss for the V1 and V2 models on MMLU and CSQA. We see that indeed the point of emergence differs between the two model series. On both benchmarks the V2 data seems to be superior, in that emergence occurs earlier, enabling greater performance earlier in pretraining.

**Prediction results.**  We now use our emergence law methodology to predict the point of emergence for both model series using only pre-emergence checkpoints. Similar, to the V1 series, the 3B checkpoints in the V2 series are all pre-emergence. Therefore, just as we did in Section 4, we hold out the larger models and only use finetune our 3B model checkpoints for fitting our emergence law. For both model series we use the same finetuning datasubsets for fitting the emergence law (see Appendix A.6). We see in Figure 9, that indeed our emergence

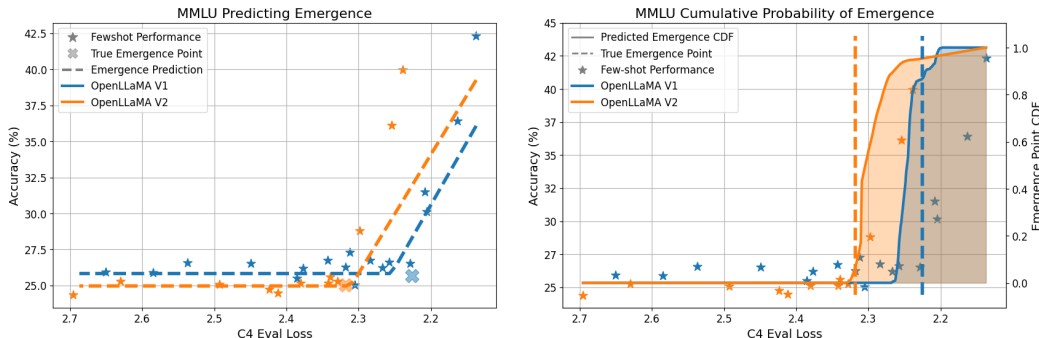

Figure 9: Comparing emergence predictions for OpenLLaMA V1 verses V2 on MMLU as a function of C4 validation loss. We plot the maximum likelihood predictions on the left and the CDFs on the right. Our emergence law predicts that the V2 models will emerge before the V1 models, suggesting that the V2 data may be of higher quality. This result provides a proof of concept demonstration for the utility of emergence prediction for enabling model developers to more cheaply make decisions on the basis of downstream capabilities.

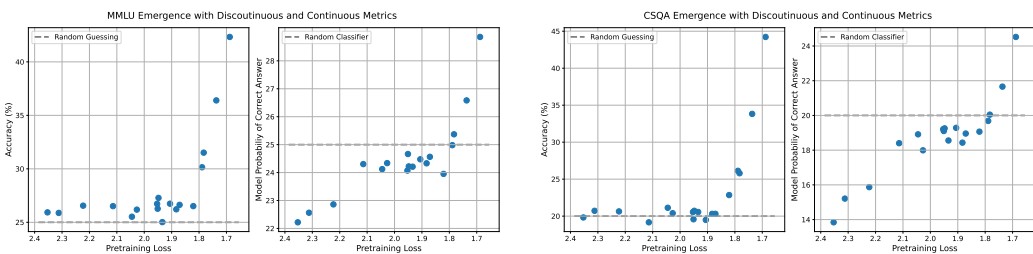

Figure 10: On a standard 5-shot MMLU and 6-shot CommonsenseQA (CSQA) evaluation, we observe emergence using both the standard correct answer accuracy evaluation and a continuous LM log-probability metric.

law predicts that OpenLLaMA V2 will emerge before OpenLLaMA V1, informing us that the V2 pretraining data may be of higher quality.

This experiment demonstrates the utility of emergence prediction for enabling model developers to more cheaply make modeling decisions on the basis of downstream performance. According to Section 4, we can potentially save up to 4x the pretraining compute by using emergence laws to more cheaply make decisions. While our emergence law does not necessarily tell us how performance will scale after the point of emergence, we believe that simply knowing that emergence will occur earlier may be a sufficient signal for making decisions in many settings. Future work should also build on our findings to enable additional post-emergence predictions (e.g. the rate of improvement post-emergence) from pre-emergence model checkpoints.

### A.3 Emergence with Continuous Metrics

In Figure 10 we present two LM tasks (MMLU and CommonsenseQA) in which we observe emergence with both continuous and discontinuous metrics.

### A.4 Comparing Emergence Law Functional Forms

In Table 1, we compare the prediction accuracy of several different emergence law functional forms. We see that a power-law in the log of the data makes better predictions than a power-law without the log. Additionally, we see that removing the optional few-shot term has little effect on prediction quality, and therefore we consider this term to be optional.

| Functional Form | GSM8K | MMLU | CSQA | CoLA |
|---|---|---|---|---|
| Log power law | 0.022 [0.004, 0.170] | **0.041 [0.011, 0.055]** | **0.003 [0.001, 0.045]** | **0.064 [0.030, 0.121]** |
| Power law | **0.013 [0.004, 0.149]** | 0.046 [0.002, 0.113] | 0.127 [0.102, 0.165] | 0.127 [0.094, 0.179] |
| No few shot | 0.030 [0.004, 0.141] | 0.060 [0.059, 0.061] | 0.004 [0.001, 0.073] | **0.064 [0.030, 0.120]** |

Table 1: Comparing the prediction error between different emergence law functional forms. "log power law" is the full emergence law, including the optional few-shot term, presented in Section 3.3. The "power law" law differs from "log power law" in that it models the shift in the emergence point using a power law in the data rather than in the log of the data. Finally, "no few shot" represents removing the extra few-shot term from the "log power law" emergence law. We present the absolute error between the maximum likelihood predicted point of emergence and the ground truth. In brackets we include the 5th and 95th percentile of prediction errors produced by our MCMC posterior sampling. We see that removing the log generally results in much worse predictions, suggesting that a power-law in the log of the data is a more natural model. We also see that the additional few-shot term has relatively little effect on our prediction quality, and therefore it should be considered optional.

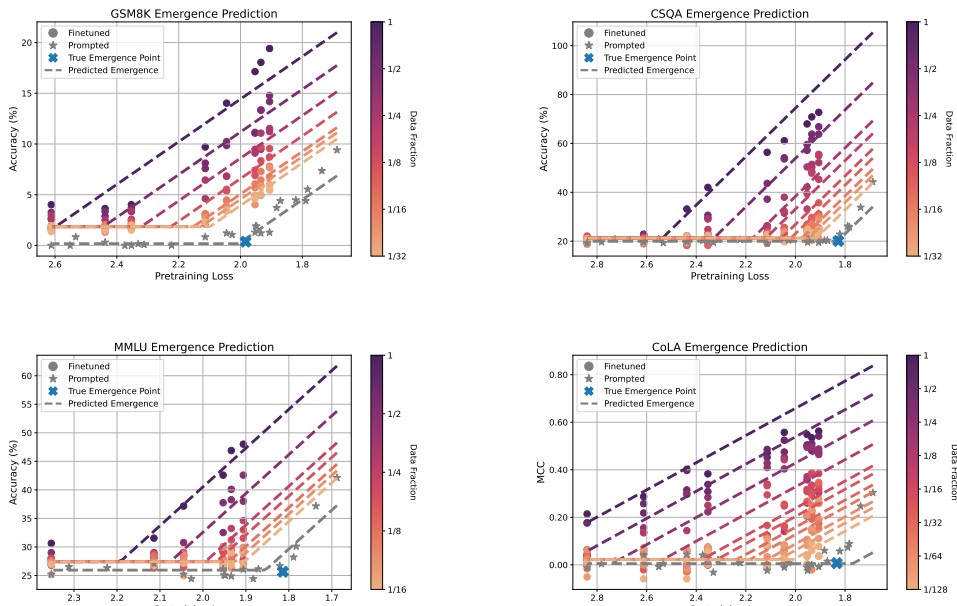

Figure 11: We plot the maximum likelihood predictions from our emergence law on each task. These plots include results from every finetuning run used for fitting the emergence law. The grey line represents our extrapolated prediction and the multi-color lines correspond to the fit produced by the emergence law for the various data levels. We see that across all tasks we are able to successfully predict the point of emergence within 0.1 nats and in many cases much less than that.

## A.5 Full Emergence Prediction Plots

In Figure 11 we plot all of the data for our maximum likelihood emergence predictions from Section 4.

## A.6 Additional Experiment Details

To select the set of finetuning data subsets used for fitting our emergence law, we first conduct a logarithmic sweep over various data amounts and then collect additional subsets nearer to the limit at which emergence is visible with our 3B checkpoints. To account for

noise in the data sub-sampling process, we finetune on two different sampled subsets for each data subset level.

For each task we perform full parameter finetuning using the AdamW optimizer. We use 0.05 dropout, 0.01 weight decay, Adam $\beta_1 = 0.9$, $\beta_2 = 0.95$, and learning rate schedule with linear warmup followed by a constant learning rate. On all experiments, we hold our 10% of our training data for validation. For each task we perform a small sweep over learning rate, batch size, and the number of learning-rate warm-up steps, selecting the best setting that produces the best validation loss. We perform early stopping according to validation loss on all tasks except on CoLA, which we describe in more detail below. For all of our experiments, except those in Appendix A.2 we use the OpenLLaMA V1 (Geng & Liu, 2023) models that were pretrained on 1T tokens from the Red-pajama dataset (Computer, 2023). In Appendix A.2, we also use the OpenLLaMA V2 models which were pretrained on a different corpus, also for ~1T tokens. Below we include more details on our experiments for each task.

**MMLU.** For predicting emergence on MMLU we use 6 intermediate 3B model checkpoints pretrained for 42B, 210B, 419B, 629B, and 1T tokens. We train on the MMLU test set and then evaluate on the validation set. In addition to the full data, we train each checkpoint on two different randomly sampled dataset subsets of each fraction: $\frac{1}{2}$, $\frac{1}{4}$, $\frac{1}{8}$, $\frac{1}{16}$, $\frac{3}{16}$, and $\frac{3}{32}$. We use learning rate 5e-6, batch size 256, and 24 learning-rate warmup steps. We use the standard 5-shot LM evaluation harness (Touvron et al., 2023) prompt for our few-shot results. We use these same settings for our experiments with OpenLLaMA V2 in Appendix A.2.

**GSM8K.** For predicting emergence on GSM8K we use 8 intermediate 3B model checkpoints pretrained for 21B, 31B, 42B, 210B, 419B, 629B, and 1T tokens. We train and test on the standard splits. On this task our models are trained to output a chain-of-thought followed by an answer. In addition to the full data, we train each checkpoint on two different randomly sampled dataset subsets of each fraction: $\frac{1}{2}$, $\frac{1}{4}$, $\frac{1}{8}$, $\frac{1}{16}$, $\frac{1}{32}$, and $\frac{3}{64}$. We use learning rate of 1e-5, a batch size of 32, and no learning-rate warmup steps. For our prompted evaluation, we use the 6-shot chain-of-thought prompt from Zelikman et al. (2022).

**CommonsenseQA.** For predicting emergence on CommonsenseQA we use 9 intermediate 3B model checkpoints pretrained for 10B, 21B, 31B, 42B, 210B, 419B, 629B, and 1T tokens. We train on the standard train set and evaluate on the validation set. In addition to the full data, we train each checkpoint on two different randomly sampled dataset subsets of each fraction: $\frac{1}{2}$, $\frac{1}{4}$, $\frac{1}{8}$, $\frac{1}{16}$, $\frac{1}{32}$, $\frac{3}{64}$, $\frac{3}{32}$, and $\frac{3}{16}$. We use a learning rate of 5e-6, a batch size of 64, and 96 learning-rate warmup steps. For our prompted evaluation we use the 7-shot prompt from Wei et al. (2022b) with the chain of thought examples stripped from the prompt, just showing the final answer.

**CoLA.** For predicting emergence on CoLA we use 9 intermediate 3B model checkpoints pretrained for 10B, 21B, 31B, 42B, 210B, 419B, 629B, and 1T tokens. We train on the standard train set and evaluate on the validation set. In addition to the full data, we train each checkpoint on two different randomly sampled dataset subsets of each fraction: $\frac{1}{2}$, $\frac{1}{4}$, $\frac{1}{8}$, $\frac{1}{16}$, $\frac{1}{32}$, $\frac{1}{64}$, $\frac{1}{128}$, $\frac{3}{64}$, $\frac{3}{128}$, and $\frac{3}{256}$. We use a learning rate of 5e-6, a batch size of 256, and 48 learning-rate warmup steps. For our prompted evaluation we use the standard 5-shot prompt in the LM evaluation harness (Touvron et al., 2023). On CoLA we deviate slightly from our realy stopping procedure. We find that for some of the smaller data subsets, downstream validation performance continues to improve after overfitting. In these cases we perform early stopping according to the downstream MCC evaluation on the validation set rather than the validation loss.

All finetuning experiments were done on TPU-V3 pods using JAX (Bradbury et al., 2018) and Scalax (Geng, 2024) for distributed training.

### A.7 Emergence Law Fitting

For obtaining our maximum-likelihood fits we follow the procedure in Hoffmann et al. (2022) and use the L-BFGS optimizer, selecting the best fit from a sweep over initializations. In particular, we first perform a brute-force grid-search over all the values in: $A \in \{0.0, 0.1, 0.2, ..., 4.0\}$, $B \in \{0.0, 0.1, 0.2, ..., 1.0\}$, $k \in \{0.0, 0.05, 0.1, ..., 1.0\}$, $\alpha \in \{1.0, 1.5, 2.0, ..., 10.0\}$, $C \in \{0.0, 0.5, 1.0, ..., 10.0\}$. We select the 100K parameters from this grid-search with the lowest loss and then run the L-BFGS optimizer initialized from each one. We then select the best fit final fit out of each of these. We fit the ReLU for the true emergence point using the same procedure.

For posterior sampling, we use the No-U-Turn Sampler Hoffman et al. (2014) implemented in Numpyro (Phan et al., 2019). We use 4 chains with 25k samples each and 15k warmup steps each, totaling 100k samples. We initialize each chain using the parameters maximum likelihood parameters found using the procedure described above. We find that without tuning the temperature of the energy function the sampler is extremely unstable, demonstrating wildly out-of-distribution posterior samples. To control this we sweep over temperature values 1e-3, 1e-4, 1e-5, 1e-6, 1e-7, 1e-8, 1e-9 and select the lowest such temperature for which the mode of the sampled distribution is centered around the maximum likelihood estimate.

