# OpenReview forum: "Predicting Emergent Capabilities by Finetuning"
_colmweb.org/COLM/2024/Conference — COLM_

### Official Review · Reviewer_Fqr2 · 2024-04-25

**Rating:** 7
**Confidence:** 4
**Ethics Flag:** 1

**Summary:**

A central question for the field, given prior work, is whether we can predict emergent capabilities. This paper present a method that shows some success: if the emergent abilities are to be elicited via prompting models, then essentially the point of emergence can be shifted to the left via finetuning models, meaning emergence (when prompting) can be predicted using models of much less compute.

1. The core idea is simple but deft. While it would be easy to dismiss the idea as quite straightforward, I see it as a a nice balance between simplicity (a good quality) and ingenuity (a good quality).
2. The predictive accuracy of the method for "emergence prediction" seems reasonable. Of course, there is nothing too convincing this is the best method or so on, but that seems fine: given this work, other papers can refine the specifics on how to best do this going forward.
3. The scale improvement of the method (i.e. how far can you shift left) seems reasonable. Similarly could be improved.

**Questions To Authors:**

1. You differentiate methods like prefix tuning from LoRA/full fine-tuning in terms of the successfulness of the left shift. Can you characterize some key properties that amount to this (e.g. is there some key property you need for this to work)? Since, if you could, this would then lead to a next question of how we search over methods with this property to identify the best method for maximizing the extent to which the emergence point gets left shited.
2. You identify two axes experimentally: the amount the point shifts to the left and the predictive accuracy. Can you characterize the relationship between these, either empirically or theoretically in some toy model? (Basically thinking about interplay between final 2 paragraphs of page 7).

**Reasons To Accept:**

1. Good idea, which if it works more generically, would be a useful insight for solving a real problem of significant scientific and societal interest (i.e. "can we predict, in advances, whether we will see emergent capabilities")
2. Results/experiments are generally interesting, reasonably executed, and provide some insight.
3. Method seems like a fine starting point, though not especially remarkable.

**Reasons To Reject:**

No major reasons to reject. I definitely don't think the paper is without flaws, but none feel so damning that I feel the need to proactively make the arguments against the paper.

---

> ### Author Rebuttal · Authors · 2024-05-30
>
> We thank reviewer Fqr225 for their detailed and well thought out comments on our work! We are glad that they agree that the problem of emergence prediction is highly important, and that they think our method strikes a good balance between simplicity and ingenuity.
>
> We answer the reviewer’s questions below:
>
> > You differentiate methods...
>
> Yes, this is indeed a fascinating line of reasoning! From the experiments we've tried so far it seems that updating the model's parameters is critical for shifting the emergence point (e.g. prefix tuning does not effect the emergence point, neither does modifying the prompt).
>
> For the camera ready, we will include some additional discussion about the potential for future work to investigate new methods which can more effectively shift the emergence point.
>
> > You identify two axes experimentally...
>
> Yes this is a good suggestion. For the camera ready, we will include ablations where we hold out finetuning runs with larger / smaller amounts of data, so as to understand empirically how this influences the accuracy of our emergence predictions.
>
> Please let us know if you have any additional comments or questions, as we believe that your feedback will help us improve the paper!

---

> > ### Comment · Reviewer_Fqr2 · 2024-06-04
> > **Reviewer response to author rebuttal**
> >
> > Thanks for the response, I have read the discussion with other reviewers as well.
> > I am keeping my score + review as-is and I recommend to the AC that this paper be accepted.

---

### Official Review · Reviewer_8WHU · 2024-04-30

**Rating:** 5
**Confidence:** 5
**Ethics Flag:** 1

**Summary:**

This paper proposed a new approach by fine-tuning LLMs for making predictions of the so-called emergent abilities. They proposed using a parametric function to predict when emergence will occur, or "emergence laws," and tested it on four NLP benchmarks where emergence has been observed. Results showed that with small-scale LLMs, they could predict emergence of models trained with more compute than the smaller models.

**Reasons To Accept:**

(1) An interesting extension to scaling law;

**Reasons To Reject:**

(1) About the motivation. Generally, I think it is less interesting to see that this work has been done based on fine-tuning data. For LLMs, it is generally useful to predict model performance during the pre-training preriod (especially the beginning) but not the fine-tuning stage. It is very challenging to make accurate prediction on task performance given limited pretraining tokens. Specially, for LLMs, it would be very strange to fine-tune it according some specific task, because it is expected to generalize on donwstream tasks.

(2) The evaluation sets. Currently, the examined tasks are widely studied or compared. It is not clear whether the approaches can generalize to more scenarios or tasks, especially difficult tasks.

(3) Background and related work seem to be overlapping. I suggest you merge these two parts.

---

> ### Author Rebuttal · Authors · 2024-05-30
>
> There may be a misunderstanding about  the role of finetuning in our paper. The goal of emergence prediction is to make predictions about the point in pretraining at which emergence in the few-shot setting will occur. In Section 3.1 we clearly describe this setting:
>
>  > We define emergence prediction as the problem of identifying the point in scaling at which emergence will occur using only checkpoints from models that are pre-emergence (i.e., have near-random or trivial performance on a task of interest). In our case, we define “model scale” to be the pretraining loss of a given model. We therefore aim to predict the point in the pretraining loss at which emergence will occur.
>
> The specific approach to emergence prediction that we propose in the paper utilizes finetuning as a tool to make such predictions. The method is not, however, designed to make predictions about finetuning performance.
>
> While we do explain this throughout the paper --- in the abstract, introduction, and in Section 3 --- such confusion is understandable and we will make sure that this is extremely clear in the camera ready version of the paper.
>
> > The evaluation…
>
> The tasks we selected were not selected due to their popularity and this has not impacted our analysis in any way; they could have been any tasks. As described in Section 4, we selected a diverse set of LM tasks on which larger, more capable models demonstrate emergence but not smaller, less capable models.
>
> To further test the generalizability of our findings with other models/architectures, we have additional finetuning experiments on the extremely challenging GPQA and APPS Interview benchmarks, using the LLaMA 2 base models (7B/13B/70B).
>
> In this setting we observe emergence with the finetuned LLaMA 2 models but random performance with all the few-shot models. With additional finetuning experiments (on-going), we will be able to fit an emergence law to the data and make a prediction about the point in scaling at which emergence would have occurred on these tasks with a hypothetical more capable model in the LLaMA 2 series. See attached figures for our current experiments with [GPQA](https://imgur.com/a/nwa7FFi) and [APPS](https://imgur.com/a/2tqLHKm).
>
> In addition to the LLaMA 2 experiments, we plan to add additional experiments with OpenLLaMA V2.
>
> > Background…
>
> Thanks! We will take this into account when preparing the camera ready.
>
> Thank you for your comments! Let us know if we can further clarify anything.

---

> > ### Comment · Reviewer_8WHU · 2024-06-03
> >
> > Partially agree with the response, and would marginally raise my score.

---

### Official Review · Reviewer_4TG4 · 2024-05-12

**Rating:** 6
**Confidence:** 2
**Ethics Flag:** 1

**Summary:**

This paper proposes a model to predict when emergence will occur in LLMs. The idea is as follows. Pretraining loss scales predictably via an inverse power law with the amount of data, compute, or parameters. However, as training loss decreases, at a certain point there is a sharp uptick (popularly termed 'emergence') in performance on certain tasks. The argument of the paper is that rather than the training loss decreasing due to more parameters, compute, or large amounts of data, the decrease in training loss associated with e.g. larger models can be simulated by fine-tuning on the given dataset. The paper proposes a model to predict when the emergence 'elbow' will occur as a function of amount of finetuning data. The paper then takes a series of models and datasets, and fits parameters for their model. It is argued that their model fits the data very well.

**Questions To Authors:**

I may be missing something here, but I didn't understand how the model is useful. I can see that it is interesting to have a model of how and when emergence occurs. However, finetuning is used as a proxy for e.g. using more parameters, and the model a function of the amount of fine-tuning data (I think). So what does this tell us? Can it tell us how to predict emergence points as models grow in size or with the use of more training data?

As the emergence point depends on different facets, could the model be extended to depend on more than one variable?

In the first equation on p.5, the quantity D is used but not explained (I guess it is the amount of fine-tuning data).

The phrase 'taking the limit ... as D approaches the number of examples in our fewshot prompt' occurs on p6. It would be really helpful to write down what the sizes of D and the few-shot prompt are.

In Fig 3 there are so many prediction lines that it is not really clear whether the predicted emergence points are realistic. Might be better to have fewer at the low data fractions, say 4 in total.

The emergence points for GSM8K seem bad, looks to me like emergence happens at around 2.2 for all amounts of data. Emergence points also look early for MMLU. This is because the RELU puts baseline performance too low, should be around 4% accuracy for data fraction 1 for GSM8K, for example, rather than 2.

Related to my first question: the model is fitted using checkpoints from OpenLlama-3B. Then, the paper states that they "then make us of the 7B and 13B checkpoints for validating the accuracy of their predictions". I couldn't see where the paper did this though.

Similarly figure 5 was hard to interpret. I understand the LMs we don't have access to are those that are trained for longer. There aren't any predictions for these points though - why are they on the plots? Why is True Emergence only one line? It looks like this corresponds to the emergence loss values in Fig 3 for each dataset, but surely the emergence loss value is going to be different for different models. Why are the failed predictions all on the True Emergence line rather than being placed at their actual locations? (I guess the x value is too far off the plot?) Why do some of the failed predictions have no corresponding LM? I also don't see how to read from the plots the reported fact that 'on MMLU and GSM8K we are able to predict emergence with up to 4x the FLOPS in advance of the emergence point'. It would be good to give a bit more guidance on how to read this. I apologise if these questions are all very basic, but I feel like the plot could be explained better.

Minor questions/typos:
- Please number equations
- Please make sure all your terms are defined.

**Reasons To Accept:**

- The topic of how and when emergence occurs in models is important

**Reasons To Reject:**

- I am not clear what the proposed model really tells us about how and when emergence occurs
- I was unclear about some of the notation and claims (details below)

---

> ### Author Rebuttal · Authors · 2024-05-30
>
> We will clarify/simplify Figure 5. While many of the reviewer’s questions are addressed in the paper, we will clarify these further with additional discussion in the final.
>
> > I may…
>
> As outlined in Sections 1/2, we believe the utility is twofold:
>
> 1) Predicting future capabilities in language models is important for safety preparedness.
> 2) To make modeling decisions based on downstream capabilities, one will need to spend much compute to see non-trivial performance, making iterating expensive/slow. We can accelerate this with predictions.
>
> > As the…
>
> Yes! In this work, we define the problem of emergence prediction and propose a particular approach, but there are many future directions (see Section 5).
>
> > In the…
>
> Yes, data amount. We will clarify this.
>
> > The phrase…
>
> Details about the prompts are in A.4; to answer: GSM8K=6-shot, MMLU=5-shot, CSQA=7-shot, CoLA=5-shot.
>
> > In Fig 3…
>
> For the final we will include plots with/without all data for completeness/clarity.
>
> > The emergence…
>
> On points corresponding to the larger data amounts, our fit may be imperfect. As explained in 3.3, we weigh the fit by the inverse data amount, to prioritize fitting in the low data limit. Our focus is on predicting the point of emergence in the limit, and our procedure is designed for this.
>
> > Related…
>
> We do this throughout. The stars in Figure 3 correspond to all models (3/7/13B), but the finetuned circles correspond only to 3B. We will clarify this.
>
> > Similarly…
>
> Figure 5 demonstrates how early we can to predict. In practice, we care about how early in FLOPs. In general, there is not a 1-to-1 mapping between loss/FLOPs. We therefore denote the loss/FLOPs for the models with stars.
>
> The “models we don’t have access to”, we do have access to (these are 7/13B models), we just don’t use them for fitting (only 3B). They are included for clarity.
>
> Yes, the failed predictions would often be off the plot. The models corresponding to some X’s are off the plot. We constrained the plot bounds for clarity. We will clarify/simplify this.
>
> To answer “Why is True Emergence only one line?”
>
> There is a single point of emergence with the OpenLLaMA-V1 series. With a different series, emergence will change (see attached for [MMLU](https://imgur.com/a/P3tyuk4) and [CSQA](https://imgur.com/a/FWKYkla)). We discuss this in Section 2; this is also covered in concurrent work ([1](https://arxiv.org/pdf/2403.15796)).
>
> Thank you for the comments! Let us know if there's anything else we can clarify.

---

> > ### Comment · Reviewer_4TG4 · 2024-06-04
> > **Response to rebuttal**
> >
> > Thanks for your responses. Could you clarify for me how to read from the plots the reported fact that 'on MMLU and GSM8K we are able to predict emergence with up to 4x the FLOPS in advance of the emergence point'.
> >
> > The other responses are fine. Please do make sure all the notation is clarified.

---

> > > ### Author Response · Authors · 2024-06-04
> > >
> > > Yes I'm happy to explain. I initially wrote up an explanation of how to interpret the plot for the initial rebuttal but had to cut it due to the character limit on the COLM OpenReview.
> > >
> > > To explain the plot with an example: for MMLU we are able to make reasonable predictions about the point of emergence using models trained with ~10^22 FLOPs, but not with fewer than 10^22 FLOPs. The first post emergence model on MMLU was trained with ~5*10^22 FLOPs, hence we are able to predict 4-5x the FLOPs in advance on this task. For GSM8K, we can do the same calculation; we take the earliest successful prediction and the first post-emergence model and the FLOPs comes out to about 4x. Though there is one failed prediction in between the successful predictions on GSM8K; this failure can likely be corrected by running additional fine-tuning runs on different data subsets, as discussed in Section 5. We will clarify this in the paper and include some additional analysis to this effect. Overall, the figure is meant provide visual clarity and detail regarding the claims in Section 4 about how far in advance we are able to make emergence predictions (e.g. 4x the FLOPs in advance / 2x the FLOPs in advance).

---

> > > > ### Comment · Reviewer_4TG4 · 2024-06-04
> > > >
> > > > Thanks, I will raise my score a little

---

### Official Review · Reviewer_GYaM · 2024-05-15

**Rating:** 6
**Confidence:** 3
**Ethics Flag:** 1

**Summary:**

The paper presents a method to predict emergent capabilities in language models by finetuning on specific tasks. Using an "emergence law" to model shifts in performance based on finetuning data, the authors successfully predict task emergence on NLP benchmarks with high accuracy, often using significantly fewer computational resources. The study offers a valuable tool for anticipating future AI capabilities and outlines areas for further research to enhance prediction methods.

**Reasons To Accept:**

- The phenomenon primarily investigated in the paper—namely, the observation that directly fine-tuning LLMs on a given task can shift the point in scaling at which emergence occurs towards less capable models—is intriguing and is supported by detailed experimental analysis.
- Explored various methods for shifting the point of emergence and provided corresponding analyses.
- Consideration of the impact of uncertainty is commendable, and the experimental results are highly reliable.

**Reasons To Reject:**

- The paper falls short in providing a thorough explanation for its subpar performance on tasks such as CommonsenseQA and CoLA, failing to convincingly demonstrate the emergence law's robustness regarding these tasks.
- The testing was limited to models within the Open-LLaMA V1 series, without indicating whether the emergence law is applicable to models of other architectures.
- The task of emergence prediction and the phenomenon primarily studied by the authors, that directly fine-tuning LLMs on a given task can shift the point in scaling at which emergence occurs towards less capable models, do not have a direct relationship, considering that the former does not involve fine-tuning. The paper lacks an explanation of the connection between these two.
- The paper fails to provide an explanation for why the emergence law curve takes a linear form.
- The paper lacks clear elucidation regarding the meanings of variables in crucial formulas. For instance, in the formula Perf(L, D) = A ∗ max(L − E(D), 0) + B, the roles of L and D are not explicitly defined.

---

> ### Author Rebuttal · Authors · 2024-05-30
>
> We address one of the main concerns about the generalizability of our findings by running additional experiments on the extremely challenging GPQA and APPS benchmarks, using the LLaMA 2 models (7B/13B/70B).
>
> In this setting we observe emergence with finetuned LLaMA 2 but random performance with few-shot. After additional finetuning experiments (on-going), we will fit an emergence law to the data and make a prediction about the point in scaling at which emergence would have occurred on these tasks with a hypothetical more capable LLaMA 2 model. See attached figures for [GPQA](https://imgur.com/a/nwa7FFi) and [APPS](https://imgur.com/a/2tqLHKm).
>
> > The paper falls...
>
> In our work we propose a challenging new problem of emergence prediction and introduce an initial approach to solving it, which makes surprisingly good predictions. However, future work can improve our results further, by refining our methodology or using new approaches for prediction. We discuss this Section 5, but will include additional discussion.
>
> > The testing...
>
> We focused on the OpenLLaMA-V1 series for the reasons outlined in Section 4, and we expect our findings to transfer to other models. To test this, we have experiments on LLaMA 2, as described above. We also will include additional experiments with OpenLLaMA-V2.
>
> > The task...
>
> In the paper, we define the general task of emergence prediction and its importance (Section 2/3). We then outline one particular approach to solving this problem, which relies on the observation that finetuning shifts the point of emergence towards less capable models (Sections 3/4). These are two separate but related contributions. We will clarify this further.
>
> > The paper fails...
>
> As discussed in 3.3, our goal with using a ReLU is to clearly model the point of emergence, rather than to accurately model performance after the point of emergence. In the latter case, alternative functional forms can be used ([1]). To model how the point of emergence shifts as a function of data, we use a power-law, which is commonly used in scaling laws ([2]).
>
> In A.2, we include ablations of variants of our functional form, giving empirical support for our decisions.
>
> [1]: https://arxiv.org/pdf/2403.08540
> [2]: https://arxiv.org/pdf/2203.15556
>
> > The paper lacks...
>
> Thanks for catching this! L is the pretraining loss and D the data amount. We will ensure this is clear in the final.
>
> Thanks for the comments! Please let us know if you have any additional thoughts.

---

### Decision · Program_Chairs · 2024-07-10

**Decision:**

Accept

**Comment:**

The paper proposes a novel approach to predict emergent capabilities in language models by utilizing fine-tuning on specific tasks, thereby anticipating the emergence of performance improvements in models with fewer computational resources. This research addresses a significant challenge in AI and presents valuable insights, supported by extensive experimental analysis. While the methodology demonstrates promising predictive accuracy, particularly in shifting the point of emergence, the paper could benefit from clearer explanations of certain variables and further validation across diverse model architectures and tasks. The reviewers appreciated the innovative approach and potential impact but highlighted areas for clarification and further validation. Overall, the paper is marginally above the acceptance threshold, with reviewers recognizing its importance and suggesting constructive improvements for the final version. Therefore, I recommend acceptance, contingent upon addressing the noted concerns and enhancing clarity.